# Wastewater-based epidemiology surveillance as an early warning system for SARS-CoV-2 in Indonesia

Indah Kartika Murni[1,2], Vicka Oktaria[1,3]*, David T. McCarthy[4,5], Endah Supriyati[6],
Titik Nuryastuti[7], Amanda Handley[8,9], Celeste M. Donato[9,10], Bayu Satria Wiratama[3],
Rizka Dinari[1], Ida Safitri Laksono[1,2], Jarir At Thobari[1], Julie E Bines[9,10,11]

**1** Center for Child Health – Pediatric Research Office, Faculty of Medicine, Public Health, and Nursing, Universitas Gadjah Mada, Yogyakarta, Indonesia, **2** Child Health Department, Faculty of Medicine, Public Health, and Nursing, Universitas Gadjah Mada, Yogyakarta, Indonesia, **3** Department of Biostatistics, Epidemiology, and Population Health, Faculty of Medicine, Public Health and Nursing, Universitas Gadjah Mada, Yogyakarta, Indonesia, **4** Department of Civil Engineering, Environmental and Public Health Microbiology Lab (EPHM Lab), Monash University, Clayton, Victoria Australia, **5** School of Civil and Environmental Engineering, Faculty of Engineering, Queensland University of Technology, Queensland, Australia, **6** Center for Tropical Medicine, Faculty of Medicine, Public Health, and Nursing, Universitas Gadjah Mada, Yogyakarta, Indonesia, **7** Department of Microbiology, Faculty of Medicine, Public Health, and Nursing, Universitas Gadjah Mada, Yogyakarta, Indonesia, **8** Medicines Development for Global Health, Southbank, Victoria Australia, **9** Enteric Diseases Group, Murdoch Children's Research Institute, Parkville, Victoria, Australia, **10** Department of Paediatrics, The University of Melbourne, Parkville, Australia, **11** Department of Gastroenterology and Clinical Nutrition, Royal Children's Hospital Melbourne, Victoria, Australia

* vicka.oktaria@ugm.ac.id

## Abstract

### Background

Wastewater-based epidemiology (WBE) surveillance has been proposed as an early warning system (EWS) for community SARS-CoV-2 transmission. However, there is limited data from low-and middle-income countries (LMICs). This study aimed to assess the ability of WBE surveillance to detect SARS-CoV-2 in formal and informal environments in Indonesia using different methods of sample collection, to compare WBE data with patterns of clinical cases of COVID-19 within the relevant communities, and to assess the WBE potential to be used as an EWS for SARS-CoV-2 outbreaks within a community.

### Materials and methods

We conducted WBE surveillance in three districts in Yogyakarta province, Indonesia, over eleven months (27 July 2021 to 7 January 2022 [Delta wave]; 18 January to 3 June 2022 [Omicron wave]). Water samples using grab, and/or passive sampling methods and soil samples were collected either weekly or fortnightly. RNA was extracted from membrane filters from processed water samples and directly from soil. Reverse-transcription quantitative real-time polymerase chain reaction (RT-qPCR) was performed to detect the SARS-CoV-2 N and ORF1ab genes.

**Data Availability Statement:** All relevant data is provided in the paper. Raw Cq data can be accessed in Wastewater Sphere (W-SPHERE,

https://sphere.waterpathogens.org/). Additional information on protocol, and steps of analysis of this study can be accessed upon request.

**Funding:** This project was funded by the Global Innovation Fund and Program for Appropriate Technology in Health (PATH). The Global Innovation Fund had no involvement in study design, data collection, or analysis while PATH participated in study design, but had no role in data collection or analysis, writing of the manuscript, or the decision to submit it for publication.

**Competing interests:** The authors have declared that no competing interests exist.

## Results

A total of 1,582 samples were collected. Detection rates of SARS-CoV-2 in wastewater reflected the incidence of community cases, with rates of 85% at the peak to 2% at the end of the Delta wave and from 94% to 11% during the Omicron wave. A 2-week lag time was observed between the detection of SARS-CoV-2 in wastewater and increasing cases in the corresponding community.

## Conclusion

WBE surveillance for SARS-CoV-2 in Indonesia was effective in monitoring patterns of cases of COVID-19 and served as an early warning system, predicting the increasing incidence of COVID-19 cases in the community.

## Introduction

Determining the burden of the Coronavirus Disease 2019 (COVID-19) in low-and middle-income countries (LMICs) is difficult and likely under-reported due to logistical barriers to detecting, tracking, and tracing the spread of SARS-CoV-2 [1].

The SARS-CoV-2 virus is present in the feces of infected people, whether or not they exhibit symptoms [2]. SARS-CoV-2 ribonucleic acid (RNA) from urine and respiratory secretions may contribute to SARS-CoV-2 in wastewater systems. Feces and other materials (sputum, urine, wash water) are transported via the wastewater collection system onto the water treatment plant. Surveillance of wastewater has been previously used to monitor select infectious agents including poliovirus, antimicrobial resistance genes, and community drug use [3]. Wastewater-based epidemiology (WBE) surveillance for SARS-CoV-2 has been proposed to monitor community transmission and as an early warning system (EWS) within the community. This could provide an opportunity to deliver and enact early, targeted public health intervention [4].

Studies reporting the detection of SARS-CoV-2 in wastewater are predominantly limited to high income countries (HICs) [5]. Few studies on the feasibility and potential benefit to the public health response in LMICs have been reported [5–8]. In this study, we aimed to assess WBE surveillance to detect SARS-CoV-2 in formal and informal environments in Indonesia. We compared WBE data with patterns of clinical cases of COVID-19 within the relevant communities and assess the potential for WBE to be used as an EWS for SARS-CoV-2 outbreaks within a community.

## Materials and methods

### Study design, setting, and populations

The SWESP study (SARS-CoV-2 surveillance using Wastewater and Environmental Sampling Project) was conducted in ten sub-districts from three districts (Yogyakarta city, Bantul, and Sleman) in Yogyakarta province. Yogyakarta province is populated by approximately 3.6 million people in urban, semi-urban, and rural communities. The urban population was represented in Yogyakarta city (373,589 people), while the semi-urban and rural populations were represented by Sleman district (1.1 million people), and Bantul district (985,770 people) respectively [9]. Six percent of the population was serviced by formal wastewater treatment

systems consisting of a central wastewater treatment plant (WWTP) or community WWTPs. Sites were selected based on population density, drainage pattern of the central and community WWTP and patterns of clinical prevalence of COVID-19 disease within the community (S1 Fig).

Samples were collected from formal sewage systems including the central WWTP and smaller community WWTPs, and from the sewage collection network accessed via manholes. Samples were also collected from major rivers that pass-through settlements in the region, from targeted high-risk sites in public facilities such as septic tanks in traditional markets and mosques, and from sites within a confined population (near source tracking [NST] sites) such as school dormitories, offices, and factories.

The SWESP study was conducted during two periods, during the peak of the 'Delta wave' between 27 July 2021 and 7 January 2022 [10] (week 1–24 of sample collection), and during the initiation of the 'Omicron wave' between 18 January to 3 June 2022 [11] (week 26–45 of sample collection, samples were not collected in week 25). The initial period during the Delta wave was a proof of concept and feasibility study (87 locations, sample n = 1,227), whereas the sample collection at the commencement of the Omicron wave was to explore if WBE surveillance had the potential to act as an EWS for the emergence of a new wave of disease in ten existing locations and eight new locations (sample n = 355). The timeline of the SWESP study and the epidemiological trend of the COVID-19 pandemic in Indonesia are presented in S2 Fig.

## Sample collection and preparation

Wastewater samples were collected using either a grab or passive sampling method as previously described [12–14].

## RNA extraction

RNA was extracted using the RNeasy PowerMicrobiome Kit (QIAGEN, Germany) following the manufacturer's instructions, with the exception of replacing the supplied beads with PowerBead Tubes-Garnet beads (QIAGEN, Germany). Extractions were spiked with 20μL of the MS2 bacteriophage internal positive control as supplied in the PerkinElmer SARS-CoV-2 Nucleic Acid Detection Kit (PerkinElmer®, USA). Eluted RNA was directly used for the Reverse-transcription Quantitative Real-time Polymerase Chain Reaction (RT-qPCR) or stored at -80˚C until RT-qPCR was performed.

## RT-qPCR and gene copy estimation

The RT-qPCR was performed using the SARS-CoV-2 Real-time RT-PCR Assay (PerkinElmer®, USA); the multiplex assay detects the nucleocapsid (N) gene and the open reading frame 1ab (ORF1ab) gene. The full process is described in detail in the S1 File.

## Epidemiological data

Weekly epidemiological data of new confirmed cases per day and deaths per day related to COVID-19 disease in Yogyakarta province were obtained from the official records of each district health office website (https://corona.slemankab.go.id/ for Sleman, https://corona.jogjakota.go.id/ for Yogyakarta, and https://dinkes.bantulkab.go.id/ for Bantul). Data of the weekly sub-district COVID-19 cases were paired with environmental SARS-CoV-2 detection data except for the central WWTP, which services all study areas where the weekly COVID-19 case was calculated based on the average of weekly COVID-19 case in all ten sub-districts.

## Data analysis

All data were analyzed using STATA 16 (STATACorp LP, College Station Texas). The positivity rate was the number of the total positive SARS-CoV-2 samples across all different types of sample sources with total samples tested in the denominator. We identified hotspots as locations that consistently have positivity rates of 60% or above of the total number of collected samples. Time-series analysis was performed to examine the weekly trend of association between SARS-CoV-2 detection in wastewater and new confirmed cases in the community. Missing values in week 25 were imputed by averaging the values in week 24 and 26 because around these weeks the Omicron cases started to increase after the flattened curve of Delta wave. As the data were not normally distributed, SARS-CoV-2 data from the environmental samples and new confirmed cases were transformed into a logarithmic scale. We calculated the exponentially weighted moving average for the log-transformed gene copies per reaction and new confirmed cases data. We adapted an analysis from the Precision Study in Bangalore, India (unpublished, personal communication) that weighted the mean average sample and cases data by 70% for the data on the observed week ($t$), and 30% for the data of the previous week ($t-1$). Data from similar sampling sources were aggregated by week and paired with new confirmed case data for the same week, and for the next one to two following weeks (1-and 2-week lag) and were further analyzed based according to sampling methods (i.e., grab [water and soil], and passive sampling [water]).

## Ethics

The study was approved by the Medical and Health Research Ethics Committee, Universitas Gadjah Mada, Indonesia (KE/FK/0426/EC/2021, KE/FK/0514/EC/2022).

## Results

### SARS-COV-2 detection in the environmental sampling in relation to laboratory-confirmed COVID-19 cases in the community

During a 45-week period, 1,582 samples (n = 1,227 collected between 27 July 2021 and 7 January 2022, and n = 355 collected between 18 January and 3 June 2022). Samples were collected on a weekly (n = 711) or fortnightly frequency (n = 871). Of 1,582 samples, 761 (48.1%) were sourced from manholes, 43 (2.7%) from central WWTP, 145 (9.2%) from community WWTP, 119 (7.5%) from soil, 398 (25.2%) from NST, and 116 (7.3%) from rivers (S1 Table).

The highest positivity rate during the Delta wave was 85.1% (1st week of sampling, 27–30 July 2021) and the lowest was 1.8% (week 23, 27–31 December 2021, Fig 1). After the flattened epidemiological curve, during the Omicron wave, the highest positivity rate was 94.4% (week 31, 21–25 February 2022) whereas the lowest was 11.1% (week 44, 23–27 May 2022). Samples sourced from manholes reached 100% positivity in the first week of Delta wave and between 7 February and 20 March 2022 (Omicron wave). Several locations consistently had positive results (Hotspots, Fig 2), including COVID-19 shelters, main manholes, central WWTP, and community WWTP.

During the Delta wave (27 July 2021–7 January 2022) and based on N gene target, 226/621 (36.4%) of wastewater samples (grab method) were positive, 94/262 (35.9%) samples of NST (passive samplers) were positive, 20/96 (20.8%) river samples (grab method) were positive, 13/24 (54.2%) central WWTP, 37/105 (35.2%) community WWTP, and 2/119 (1.7%) soil samples were positive (Table 1). The finding was consistent with the ORF1ab gene, with the exception being a higher proportion of soil samples were positive (5/119, 4.2%) for ORF1ab gene. During the Omicron wave (18 January-3 June 2022), we found a similar positivity trend. A high

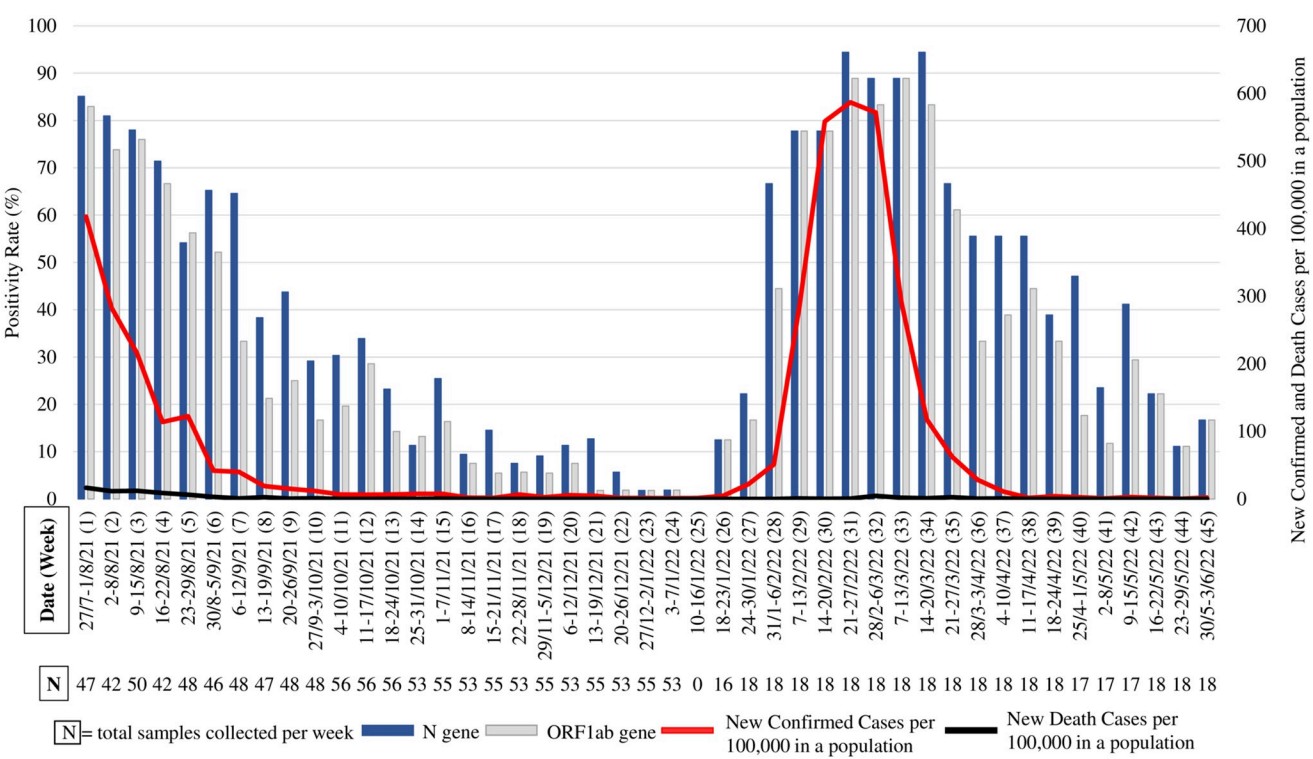

**Fig 1. Proportion of samples from all sources tested positive for SARS-CoV-2 during Delta and Omicron wave.** Samples were taken during Delta wave, week 1–24 (27 July 2021–7 January 2022) and Omicron wave, week 26–45 (18 January-3 June 2022). Samples were not collected in week 25 (10–16 January 2022).

correlation was observed between N gene and ORF1ab gene in almost all sample sources, both during the Delta and Omicron waves (Table 1). Further, N and ORF1ab genes collected from manholes showed the highest correlation (*r* = 0.972, *P*<0.001, S3 Fig).

## Wastewater-based epidemiology surveillance's potential to act as an early warning system

The weekly trend of viral load detected in the sewage mimicked the weekly new confirmed-cases of COVID-19 within the same week for grab samples collected from manholes (Fig 3a) and passive samplers (Fig 3b).

Further, we considered the relationship between wastewater viral load after adjustment for recovery using a 1-and 2-week lag analysis, to explore the potential of WBE as an EWS for COVID-19. Viral load in grab samples obtained from manholes preceded the increasing trend of cases in the community by one and two weeks (Fig 3c–3f).

## Discussion

Wastewater-based epidemiology has faced challenges in LMICs due to limited formal sewerage systems. The identification of SARS-CoV-2 in wastewater samples one and two weeks prior to the emergence of a spike of COVID-19 cases suggests that WBE has the potential to act as an EWS for detection of an emerging COVID-19 outbreak in an LMIC setting. This supports the potential for WBE surveillance in Indonesia to monitor transmission within the community and estimate changes at a population level. This is consistent with reports from other LMICs

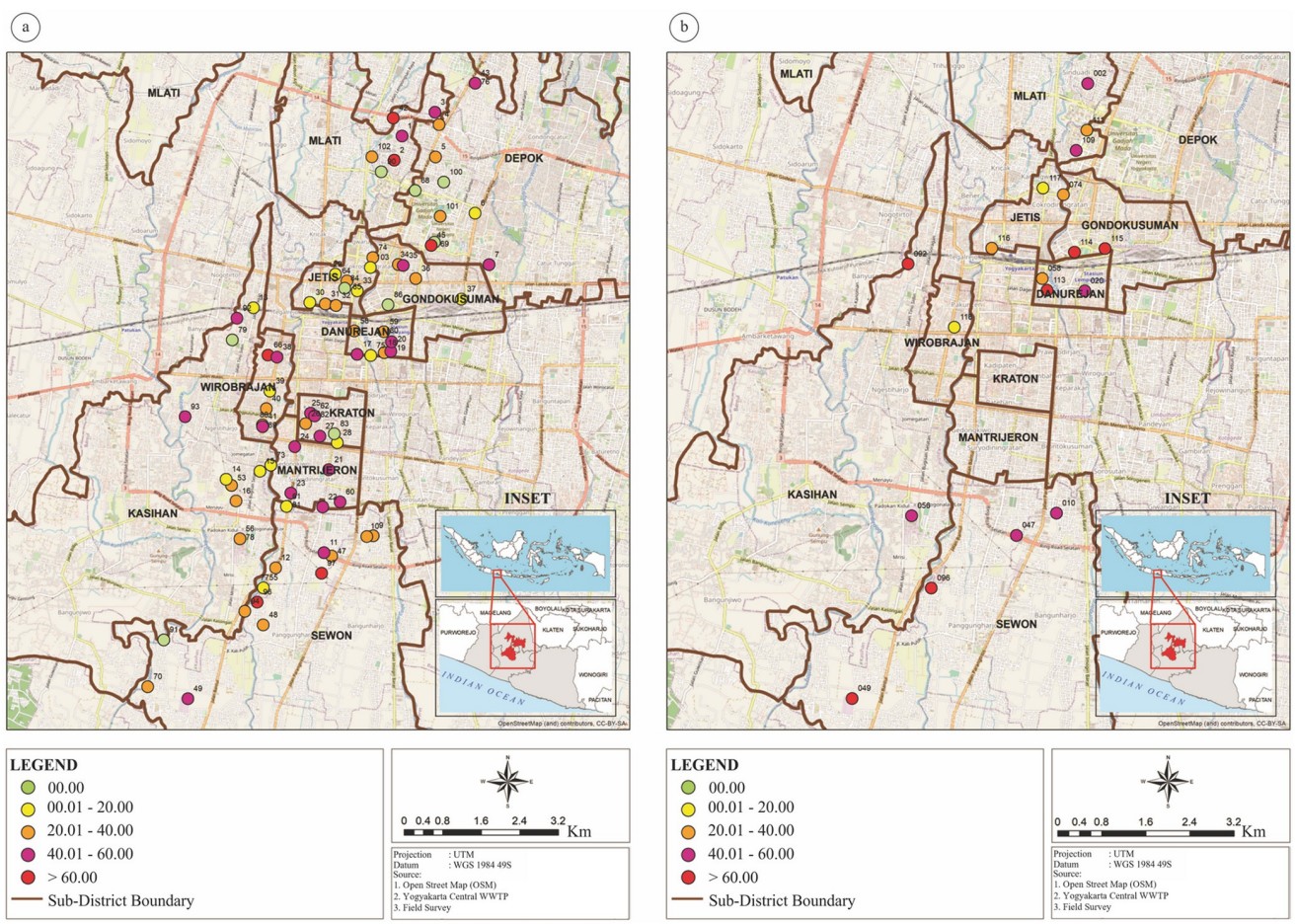

**Fig 2. Hotspot location based on the period of sample collection.** 4a. During Delta wave between 27 July 2021 and 7 January 2022 and during. 4b. During the Omicron wave between 18 January and 3 June 2022. The classification of hotspots was defined based on the percentage of positive cases in each sampling location adjusted to the sampling frequency in each sampling location (compared to the total weeks of sampling in each Delta and Omicron wave).

[5–7, 15, 16]. This study is notable due to the large sample size using different sample collection methods, in a range of high-risk locations, over a prolonged period during two distinct SARS-CoV-2 variant waves. Despite the widespread use of WBE for monitoring COVID-19, debate continues around the accuracy and reliability of the viral concentration data from wastewater to estimate the prevalence of SARS-CoV-2 infection in the community. Factors that may affect SARS-CoV-2 concentration in wastewater systems include integrity of the sewerage pipe infrastructures and rainfall [17]. Further, the ability to use the SARS-CoV-2 viral load in wastewater samples to inform the community burden of the infection has also posed challenges due to the potential confounding variables including population mobility, variability of shedding between individuals, and personal habits [4, 13].

We explored if WBE conducted in high-risk areas, such as factories, schools, or university campuses, might provide an opportunity to inform a targeted public health response. For example, during 2021 COVID-19 outbreaks in Islamic Boarding Schools were a major focus of public concern, resulting in school closures. Monitoring of infection risk using WBE at the school level has the potential to provide reassurance to the community to support face-to-face

**Table 1.  SARS-CoV-2 positive detected using N and ORF1ab genes in each sample source and its correlation.**

| Sample source | N | Number of SARS-CoV-2 positive, No (%) | | Gene copies per reaction*, Median (IQR) | | |
|---|---|---|---|---|---|---|
| | | N gene | ORF1ab gene | N gene | ORF1ab gene | r (P-value)# |
| A. Delta wave period: 27 July 2021–7 January 2022 | | | | | | |
| Manhole (grab) | 621 | 226 (36.4%) | 182 (29.3%) | 2.3 (0.99–10.4) | 1.8 (0.8–5.5) | 0.926 (<0.001) |
| Central WWTP (passive) | 24 | 13 (54.2%) | 13 (54.2%) | 145.6 (47.9–373.1) | 92.8 (46.9–269.8) | 0.877 (<0.001) |
| Community WWTP (passive) | 105 | 37 (35.2%) | 25 (23.8%) | 54.2 (25.0–189.6) | 39.2 (25.0–110.1) | 0.836 (<0.001) |
| Soil | 119 | 2 (1.7%) | 5 (4.2%) | 100 (100.0–100.0) | 100 (100.0–100.0) | 0.863 (<0.001) |
| NST (passive) | 262 | 94 (35.9%) | 68 (25.95%) | 51.6 (25.0–166.7) | 39.3 (25.0–106.2) | 0.836 (<0.001) |
| River (grab) | 96 | 20 (20.8%) | 12 (12.5%) | 0.05 (0.03–0.1) | 0.04 (0.03–0.09) | 0.906 (<0.001) |
| Total | 1227 | 392 (31.9%) | 305 (24.8%) | 25 (1.33–100.0) | 14.09 (1.11–83.9) | 0.969 (<0.001) |
| B. Omicron wave period: 18 January-3 June 2022 | | | | | | |
| Manhole (grab) | 140 | 87 (62.1%) | 77 (55.0%) | 7.1 (2.2–174.1) | 4.6 (1.8–79.3) | 0.967 (<0.001) |
| Central WWTP (passive) | 19 | 17 (89.5%) | 15 (78.95%) | 1784.23 (255.2–8909.8) | 308.7 (159.5–2838.8) | 0.939 (<0.001) |
| Community WWTP (passive) | 40 | 30 (75.0%) | 27 (67.5%) | 532.6 (78.8–2073.8) | 160.2 (42.2–764.4) | 0.933 (<0.001) |
| NST (passive) | 136 | 48 (35.3%) | 34 (25.0%) | 66.4 (28.0–160.7) | 51.4 (25.0–128.5) | 0.919 (<0.001) |
| River (grab) | 20 | 7 (35.0%) | 7 (35.0%) | 0.08 (0.03–1.2) | 0.08 (0.03–0.2) | 0.989 (<0.001) |
| Total | 355 | 189 (53.2%) | 160 (45.1%) | 52.03 (6.6–427.9) | 34.29 (4.4–173.5) | 0.971 (<0.001) |

WWTP, wastewater treatment plant; NST, near-source tracking.

* recovery adjusted

# Spearman's correlation test for N and ORF1ab genes agreement.

learning. WBE surveillance using NST provides public health officials with insight into the spread of COVID-19 within individual populations where rapid response can reduce the risk of much larger outbreaks.

Grab samples from manholes appear to provide reliable detection of SARS-CoV-2 which was highly correlated with clinical cases of COVID-19. However, passive samplers were found to be easy to deploy, suitable for sampling in low-flow sites, and avoided the collection of large volumes of water that required filtration in the laboratory [12, 14]. A disadvantage of the large-scale use of passive samplers in an LMIC is the need for access to a 3D printer to manufacture. Grab sampling is a cheap and rapid method of collection, yet it represents sampling at a single time point and therefore is subject to hour-to-hour and day-to-day variation in wastewater flow and composition. To minimize the risk of sampling variability, we collected samples at the same time in the morning when the peak load occurred in almost all sampling points. This study is among the first to report SARS-CoV-2 contamination of the soil in high-traffic public spaces such as around a factory, city square, and traditional markets, although the positivity rate in soil samples was low.

Wastewater is a complex mix of biological and chemical factors that may inhibit the quantification of SARS-CoV-2 genes, complicating data interpretation. We have reported the recovery efficiency of the samples to compensate for this possibility. The ability to accurately represent time-averaged viral concentration in wastewater needs to be validated as non-linear uptake may occur when passive samplers have reached their capacity during their exposure period so adsorption and desorption rates may be similar [18]. In addition to monitoring the community burden of infection, WBE surveillance has the potential to act as a system for the early detection of a SARS-CoV-2 outbreak within a community. This current study showed that there was approximately a 1-week lag time between the detection of SARS-CoV-2 in wastewater samples

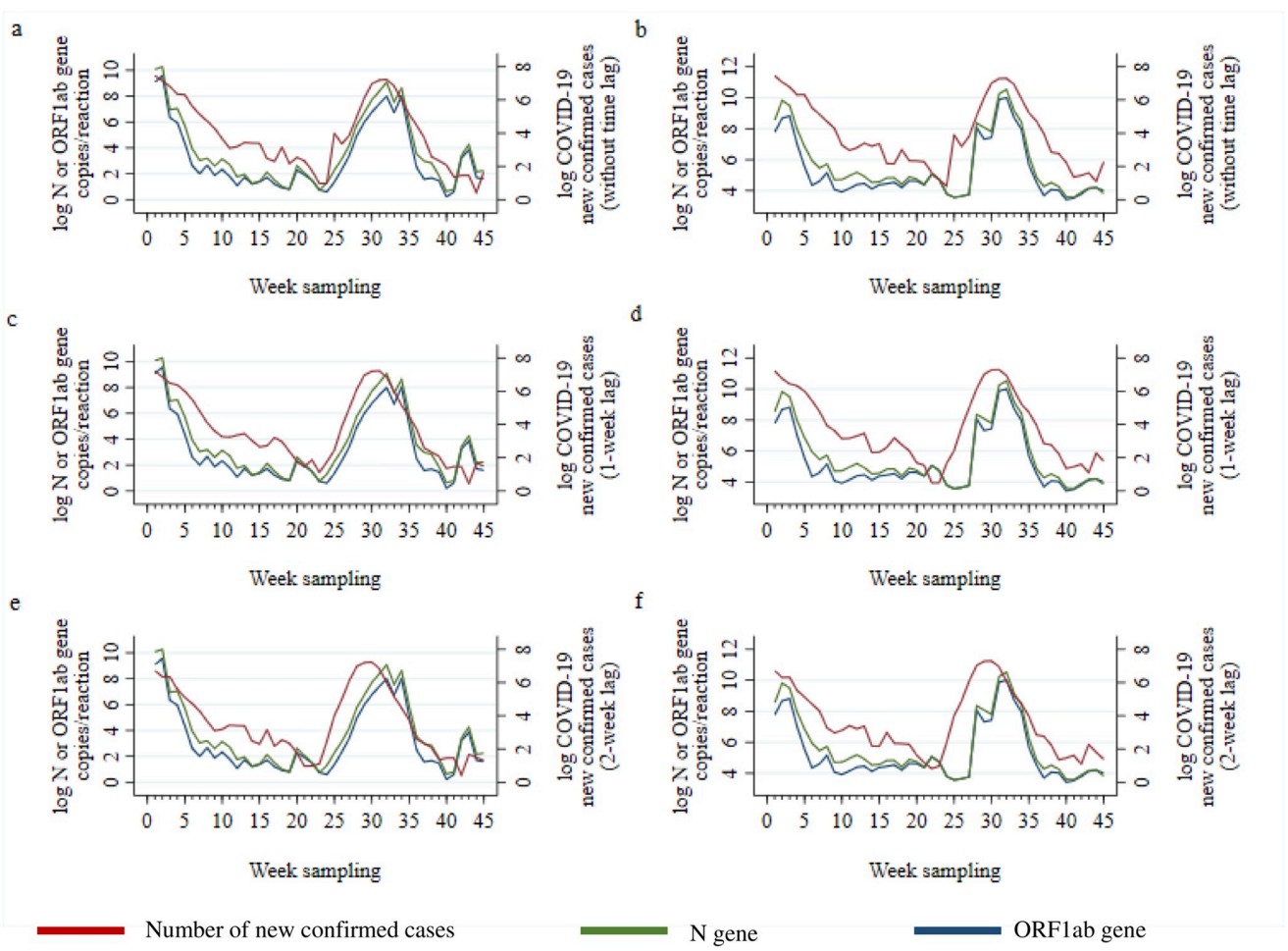

**Fig 3. COVID-19 wastewater surveillance N and ORF1ab genes against new confirmed cases after recovery adjustment by week.** Manhole sites collected by grab sampling (3a, 3c, 3e). NST water collected by passive sampling (3b, 3d, 3f). 3a. correlation between N gene and ORF1ab vs new confirmed cases were 0.808 ($P<0.001$), and 0.785 ($P<0.001$), without lag time. 3b. correlation between N gene and ORF1ab vs new confirmed cases were 0.763, ($P<0.001$), and 0.646 ($P<0.001$), without lag time. 3c correlation between N gene and ORF1ab vs new confirmed cases were 0.776, ($P<0.001$), and 0.760 ($P<0.001$), with 1-week lag time. 3d. correlation between N gene and ORF1ab vs new confirmed cases were 0.778, ($P<0.001$), and 0.658 ($P<0.001$), with 1-week lag time. 3e correlation between N gene and ORF1ab vs new confirmed cases were 0.754, ($P<0.001$), and 0.715 ($P<0.001$), with 2-week lag time. 3f. correlation between N gene and ORF1ab vs new confirmed cases were 0.671, ($P<0.001$), and 0.558 ($P<0.001$), with 2-week lag time. Red lines represent new confirmed cases, green lines represent N gene, and blue lines represent ORF1ab gene.

and the recognition of an increase in community COVID-19 cases. We observed that the epidemiological curve for COVID-19 cases was wider during Delta wave compared to Omicron wave the gap between the rate of increase of viral concentration in wastewater samples and community COVID-19 cases was narrower during the Omicron wave, perhaps due to a higher reproductive rate number (2.5 to 3.8 in Omicron vs Delta) [19]. This observation is consistent with other studies in which viral concentration of SARS-CoV-2 from wastewater correlated to the new weekly COVID-19 cases and the rolling 7-day average of active cases in the subsequent week [20]. This time lag could provide a critical window of opportunity to implement public health actions, including policy adjustment and resource planning, allocation, and preparedness as a preventive measure to immediate surge of cases in the community. In countries such as Bangladesh or Australia [21], the WBE surveillance has been integrated within the public health surveillance system and being used by stakeholders for policy making.

The implementation of a routine WBE surveillance for SARS-CoV-2 program, requires multi-stakeholder collaboration among academics, healthcare professionals, public health officers, local government, WWTP personnel, risk communicators, and the community. This will need clear frameworks for the roles of each entity. The COVID-19 pandemic has demonstrated that WBE surveillance has the potential as a valuable surveillance tool for broader applications including future pandemic preparedness for SARS-CoV-2, disease burden measurement, evaluation of public health intervention, and outbreak investigation for pathogens such as measles, influenza, and polio. However, not all data generated from WBE have to be reported in a real-time manner, depending on the pathogens, purpose, and nature of the surveillance. For instance, WBE surveillance for antimicrobial resistance does not require real-time or frequent sampling. Other considerations needed for implementation include sampling strategies that need to be tailored to Indonesian infrastructure. As the coverage for central WWTP is limited, in order to cover a wider community, we also conducted sampling not only in central WWTP but also in communal WWTPs and using the NST approach such as public spaces, offices, schools, and residential areas. Therefore, the country-specific public health characteristics in terms of pathogens and infrastructures need to be considered.

The current national implementation of WBE surveillance in Indonesia focuses on poliovirus, initiated in 2017 as a complement to Acute Flaccid Paralysis surveillance, supporting the global polio eradication program. This surveillance aids in tracking disease transmission and informing vaccination strategies during the 2022 polio outbreaks by monitoring the circulation of wild-type poliovirus and vaccine-derived poliovirus (VDPV). Further implementation of WBE includes SARS-CoV-2 surveillance, as the Indonesian Ministry of Health integrates WBE into the national surveillance system as part of a post-pandemic transition plan to monitor emerging variants.

There are some limitations of this study. As this study was conducted in one province in Indonesia, the trends in SARS-CoV-2 detection from wastewater may not be generalizable across Indonesia. We have tried to limit variability within the sites studied by collecting samples from a range of urban, semi-urban, and rural sites to gain insight into any regional differences. A practical limitation of WBE in the current study is that results were not available in real-time and required a 5–8 day period between sample collection and data reporting. If this WBE system is to be established as an EWS, the process would need to be streamlined so that data is available to best inform a timely public health response. This would require sustainable funding to support the availability of essential reagents and equipment as well as dedicated skilled personnel.

## Conclusions

WBE surveillance was effective in detecting SARS-CoV-2 in wastewater and environmental samples in urban, semi-urban, and rural areas in Yogyakarta. SARS-CoV-2 was detected in wastewater one to two weeks prior to the emergence of clinical cases in the community which supports the use of WBE surveillance as an EWS for predicting community SARS-CoV-2 transmission. This study provides proof of concept of WBE surveillance to provide data on the community disease burden with the potential to provide early warning of outbreaks of SARS-CoV-2 in an LMIC setting with a limited formal sewerage system to assist a targeted public health response in a pandemic.

## Supporting information

**S1 Fig. SWESP study sampling locations.** During the Delta wave, locations included manholes (n = 40), permanent residences (n = 3), temporary residences (n = 3), public facilities

(n = 20), offices (n = 3), rivers (n = 6), WWTPs (n = 12). During the Omicron wave, locations included manholes (n = 7), permanent residences (n = 2), offices (n = 2), schools (n = 3), rivers (n = 1), WWTPs (n = 3). All manholes were connected to the central WWTPs.
(TIF)

**S2 Fig. The timeline of the SWESP studies during the Delta and Omicron waves.** Source: https://ourworldindata.org/covid-vaccinations?country=IDN, accessed 27 July 2022.
(TIF)

**S3 Fig. Correlation in all manhole sites throughout the study, regardless of the period of sample collection, between N gene and ORF1ab gene.** Recovery adjusted, $r = 0.972$, $P<0.001$.
(TIF)

**S1 File. Extraction and RT-qPCR protocol.**
(DOCX)

**S1 Table. Summary of sample sources.**
(DOCX)

# Acknowledgments

We would like to thank the wastewater treatment plant Yogyakarta (Balai PIALAM) team, field assistants, and laboratory team for doing sampling collection and laboratory works. We thank Dwi Astuti Dharma Putri, Hendri Marinda Sari, Bayu Adji Pratama, and Alfian Yoga Sulistya for their assistance in data collection and study conduct.

# Author Contributions

**Conceptualization:** Indah Kartika Murni, Vicka Oktaria, David T. McCarthy, Amanda Handley, Celeste M. Donato, Julie E Bines.

**Data curation:** Endah Supriyati, Amanda Handley, Rizka Dinari.

**Formal analysis:** Indah Kartika Murni, Vicka Oktaria, Bayu Satria Wiratama, Rizka Dinari.

**Funding acquisition:** Amanda Handley, Julie E Bines.

**Investigation:** Indah Kartika Murni, Vicka Oktaria, David T. McCarthy, Amanda Handley, Celeste M. Donato, Julie E Bines.

**Methodology:** Indah Kartika Murni, Vicka Oktaria, David T. McCarthy, Endah Supriyati, Titik Nuryastuti, Amanda Handley, Celeste M. Donato, Bayu Satria Wiratama, Julie E Bines.

**Project administration:** Endah Supriyati, Amanda Handley, Rizka Dinari.

**Resources:** Indah Kartika Murni, Vicka Oktaria, David T. McCarthy, Titik Nuryastuti, Julie E Bines.

**Supervision:** Indah Kartika Murni, Vicka Oktaria, Titik Nuryastuti, Ida Safitri Laksono, Jarir At Thobari, Julie E Bines.

**Validation:** Indah Kartika Murni, Vicka Oktaria, David T. McCarthy, Endah Supriyati, Titik Nuryastuti, Amanda Handley, Julie E Bines.

**Writing – original draft:** Indah Kartika Murni.

**Writing – review & editing:** Indah Kartika Murni, Vicka Oktaria, David T. McCarthy, Endah Supriyati, Titik Nuryastuti, Amanda Handley, Celeste M. Donato, Ida Safitri Laksono, Jarir At Thobari, Julie E Bines.

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
