## [Decision Letter · Decision Letter 0]

21 May 2024

PONE-D-24-06901Wastewater-based epidemiology surveillance as an early warning system for SARS-CoV-2 in IndonesiaPLOS ONE

Dear Dr. Oktaria,

Thank you for submitting your manuscript to PLOS ONE. After careful consideration, we feel that it has merit but does not fully meet PLOS ONE’s publication criteria as it currently stands. Therefore, we invite you to submit a revised version of the manuscript that addresses the points raised during the review process.

We look forward to receiving your revised manuscript.

Kind regards,

Eleonora Nicolai, PhD

Academic Editor

PLOS ONE

2. We note that Figures 2 and S1 in your submission contain [map/satellite] images which may be copyrighted. All PLOS content is published under the Creative Commons Attribution License (CC BY 4.0), which means that the manuscript, images, and Supporting Information files will be freely available online, and any third party is permitted to access, download, copy, distribute, and use these materials in any way, even commercially, with proper attribution. For these reasons, we cannot publish previously copyrighted maps or satellite images created using proprietary data, such as Google software (Google Maps, Street View, and Earth). For more information, see our copyright guidelines: http://journals.plos.org/plosone/s/licenses-and-copyright.

a. You may seek permission from the original copyright holder of Figures 2 and S1 to publish the content specifically under the CC BY 4.0 license. 

**Comments to the Author**

1. Is the manuscript technically sound, and do the data support the conclusions?

Reviewer #1: Yes

Reviewer #2: Yes

2. Has the statistical analysis been performed appropriately and rigorously? 

Reviewer #1: Yes

Reviewer #2: Yes

3. Have the authors made all data underlying the findings in their manuscript fully available?

Reviewer #1: Yes

Reviewer #2: Yes

4. Is the manuscript presented in an intelligible fashion and written in standard English?

Reviewer #1: Yes

Reviewer #2: Yes

5. Review Comments to the Author

Reviewer #1: Revision manuscript “Wastewater-based epidemiology surveillance as an early warning system for SARS-COV-2 in Indonesia”

In this manuscript, the authors claim that wastewater-based epidemiology could be used as an early warning system for SARS-CoV-2 spread among the population. Although the study is well conducted and sound, I do not think that this approach would be realistically applicable in a near future and able to provide data in real-time so that health authorities can immediately act to prevent the spread of infection among the population. Furthermore, data need to be validated as well as the methods used.

Anyway, this type of investigation is suitable for collection of epidemiological data.

Minor point:

-Abstract, material and methods:

Line 38, the period of observation is longer than 9 months as stated in the text.

Reviewer #2: This is a study on the monitoring of SARS-CoV-2 in municipal wastewater samples in a specific region of Indonesia. The study was well-conducted, and the results provide important information to support epidemiological surveillance in the region, especially where the capacity for clinical testing is limited. The WBE tool has proven to be efficient and an important aid for epidemiological surveillance, particularly in developing countries. Regarding the manuscript, although it is methodologically correct and presents important results, it does not bring any novelty or significant contribution to the advancement of knowledge in this area. Many studies in the literature address the use of WBE for this purpose. I believe the authors can significantly improve the article by detailing the sampling of wastewater in the region, discussing the challenges encountered and the solutions adopted to overcome them. Additionally, they could discuss how the results were utilized by local or governmental agencies; if they were not used, they should highlight these needs and how they could be addressed. To further enhance the text, the authors could provide some examples of WBE applications in situations similar to that of Indonesia, particularly relating to issues of basic sanitation infrastructure and vulnerability indices of the monitored population.

6. PLOS authors have the option to publish the peer review history of their article (what does this mean?). If published, this will include your full peer review and any attached files.

Reviewer #1: No

Reviewer #2: **Yes: **Rodrigo de Freitas Bueno

---

## [Author Response · Author response to Decision Letter 0]

25 Jun 2024

Dear Eleonora Nicolai, PhD

Academic Editor

PLOS ONE

Thank you for inviting us to submit a revised draft of our manuscript entitled, “Wastewater-based epidemiology surveillance as an early warning system for SARS-COV-2 in Indonesia” to PLOS ONE. We also appreciate the time and effort you and each of the reviewers have dedicated to providing insightful feedback on ways to strengthen our paper. Thus, it is with great pleasure that we resubmit our article for further consideration. We have incorporated changes that reflect the detailed suggestions you have graciously provided. We also hope that our edits and the responses we provide below satisfactorily address all the issues and concerns you and the reviewers have noted.

To facilitate your review of our revisions, the following is a point-by-point response to the questions and comments delivered in your letter dated 22 May 2024. 

Feedback#1

Please ensure that your manuscript meets PLOS ONE's style requirements, including those for file naming

Response#1

Thank you for the reminder. Our file naming and authors’ and their affiliations have met the PLOS ONE’s style format. We revised new affiliations to authors named David McCarthy (line 16-17).

Feedback#2

Figures 2 and S1 in the submission contain [map/satellite] images which may be copyrighted. All PLOS content is published under the Creative Commons Attribution License (CC BY 4.0), which means that the manuscript, images, and Supporting Information files will be freely available online, and any third party is permitted to access, download, copy, distribute, and use these materials in any way, even commercially, with proper attribution. For these reasons, we cannot publish previously copyrighted maps or satellite images created using proprietary data, such as Google software (Google Maps, Street View, and Earth). For more information, see our copyright guidelines: http://journals.plos.org/plosone/s/licenses-and-copyright. We require you to either (1) present written permission from the copyright holder to publish these figures specifically under the CC BY 4.0 license, or (2) remove the figures from your submission

Response#2

Thank you for your suggestion. Regarding our Figures 2 and S1 copyright, we use free copyright sources from OpenStreetMap (OSM) website as mentioned in https://www.openstreetmap.org/copyright. We credited OSM in our images in the lower right corner. You can access the map and satellite image from www.openstreetmap.org

Feedback#3

From Reviewer #1: Revision manuscript “Wastewater-based epidemiology surveillance as an early warning system for SARS-COV-2 in Indonesia”

In this manuscript, the authors claim that wastewater-based epidemiology could be used as an early warning system for SARS-CoV-2 spread among the population. Although the study is well conducted and sound, I do not think that this approach would be realistically applicable in a near future and able to provide data in real-time so that health authorities can immediately act to prevent the spread of infection among the population. Furthermore, data need to be validated as well as the methods used.

Anyway, this type of investigation is suitable for collection of epidemiological data.

Response#3:

Not all data generated from WBE have to be reported in a real time manner, depending on the purpose and nature of the surveillance (line 309-310).

We also added the WBE utilization for assessing public health framework: (1) future pandemic preparedness, (2) measuring burden of diseases, (3) public health intervention evaluation, and (4) outbreak investigation to address the urgency and positive impact if the data provided is not in real-time (line 305-309).

Feedback#4:

Minor point: Abstract, material and methods line 38, the period of observation is longer than 9 months as stated in the text.

Response#4:

Thank you for your suggestion. We revised the period of observation from 9 months to 11 months (line 42).

Feedback#5:

From Reviewer #2: This is a study on the monitoring of SARS-CoV-2 in municipal wastewater samples in a specific region of Indonesia. The study was well-conducted, and the results provide important information to support epidemiological surveillance in the region, especially where the capacity for clinical testing is limited. The WBE tool has proven to be efficient and an important aid for epidemiological surveillance, particularly in developing countries. Regarding the manuscript, although it is methodologically correct and presents important results, it does not bring any novelty or significant contribution to the advancement of knowledge in this area. Many studies in the literature address the use of WBE for this purpose. I believe the authors can significantly improve the article by detailing the sampling of wastewater in the region, discussing the challenges encountered and the solutions adopted to overcome them. Additionally, they could discuss how the results were utilized by local or governmental agencies; if they were not used, they should highlight these needs and how they could be addressed. To further enhance the text, the authors could provide some examples of WBE applications in situations similar to that of Indonesia, particularly relating to issues of basic sanitation infrastructure and vulnerability indices of the monitored population.

Response#5:

1. We detailed the sampling of wastewater in the region by not not only covering certain regions which are covered by central WWTP, we also collected samples from communal WWTPs and NSTs (line 312-317).

2. To discuss the challenges encountered and the solutions adopted to overcome them, we outlined:

a. Challenge: trends in SARS-CoV-2 detection from wastewater may not be generalizable across Indonesia (in line 326-328), with solution: to limit variability within the sites studied by collecting samples from a range of urban, semi-urban, and rural sites to gain insight into any regional differences (line 328-330).

b. Challenge: data was not available in real-time and required a 5-8 day period between sample collection and data reporting (line 330-331), with solution: a clear framework for each stakeholder role (line 304-305), not all data generated from WBE have to be reported in a real time manner, depending on the purpose and nature of the surveillance (line 309-310) and data streamlining (line 330-333).

3. To explain on how the results were utilized by local or governmental agencies, we added that since 2017, there is an ongoing polio environmental surveillance in Indonesia, which informs vaccination strategies during the 2022 polio outbreaks. Further implementation of WBE includes SARS-CoV-2 surveillance, as the Indonesian Ministry of Health integrates WBE into the national surveillance system as part of a post-pandemic transition plan to monitor emerging variants (line 318-325).

4. Responding to 'To further enhance the text, the authors could provide some examples of WBE applications in situations similar to that of Indonesia, particularly relating to issues of basic sanitation infrastructure and vulnerability indices of the monitored population', we added explanation 'other considerations needed for implementation include sampling strategies that need to be tailored with Indonesian infrastructure. As the coverage for central WWTP is limited, in order to cover a wider community, we also conducted sampling not only in central WWTP but also in communal WWTPs and using the NST approach such as in communal WWTPs, public spaces, offices, schools, and residential areas. Therefore, the country-specific public health characteristics in terms of pathogens and infrastructures need to be considered.' (line 312-317).

Again, thank you for giving us the opportunity to strengthen our manuscript with your valuable comments and queries. We have worked hard to incorporate your feedback and hope that these revisions persuade you to accept our submission.

Sincerely,

Vicka Oktaria

Corresponding Author

vicka.oktaria@ugm.ac.id (VO)

---

## [Editor Report · Decision Letter 1]

4 Jul 2024

Wastewater-based epidemiology surveillance as an early warning system for SARS-CoV-2 in Indonesia

PONE-D-24-06901R1

Dear Dr. Vicka Oktaria,

We’re pleased to inform you that your manuscript has been judged scientifically suitable for publication and will be formally accepted for publication once it meets all outstanding technical requirements.

Kind regards,

Eleonora Nicolai, PhD

Academic Editor

PLOS ONE

Additional Editor Comments (optional):

Dear authors,

I appreciate your work in addressing all reviewers , in my opinion it gives more strength to your paper.
---

## [Editor Report · Acceptance letter]

9 Jul 2024

PONE-D-24-06901R1 

PLOS ONE

Dear Dr. Oktaria, 

I'm pleased to inform you that your manuscript has been deemed suitable for publication in PLOS ONE. Congratulations! Your manuscript is now being handed over to our production team.

Kind regards, 

on behalf of

Dr. Eleonora Nicolai 

Academic Editor

PLOS ONE